# Portable Gait Lab: Estimating Over-Ground 3D Ground Reaction Forces Using Only a Pelvis IMU

**DOI:** 10.3390/s20216363

**Published:** 2020-11-07

**Authors:** Mohamed Irfan Mohamed Refai, Bert-Jan F. van Beijnum, Jaap H. Buurke, Peter H. Veltink

**Affiliations:** 1Biomedical Signals and Systems, University of Twente, 7522NB Enschede, The Netherlands; b.j.f.vanbeijnum@utwente.nl (B.-J.F.v.B.); j.buurke@rrd.nl (J.H.B.); p.h.veltink@utwente.nl (P.H.V.); 2Roessingh Research and Development, 7522AH Enschede, The Netherlands

**Keywords:** inertial measurement units, ground reaction forces, ambulatory, minimal sensing, wearables

## Abstract

As an alternative to force plates, an inertial measurement unit (IMU) at the pelvis can offer an ambulatory method for measuring total center of mass (CoM) accelerations and, thereby, the ground reaction forces (GRF) during gait. The challenge here is to estimate the 3D components of the GRF. We employ a calibration procedure and an error state extended Kalman filter based on an earlier work to estimate the instantaneous 3D GRF for different over-ground walking patterns. The GRF were then expressed in a body-centric reference frame, to enable an ambulatory setup not related to a fixed global frame. The results were validated with ForceShoes^TM^, and the average error in estimating instantaneous shear GRF was 5.2 ± 0.5% of body weight across different variable over-ground walking tasks. The study shows that a single pelvis IMU can measure 3D GRF in a minimal and ambulatory manner during over-ground gait.

## 1. Introduction

Measuring kinetics of gait such as 3D ground reaction forces includes estimating the vertical and shear forces acting on the body during gait. The total ground reaction forces (GRF) acting on the body, and its derived parameters related to the center of mass (CoM) such as dynamic balance and stability measures [1], can be helpful in understanding gait quality [2]. Therefore, measuring GRF is useful in studying healthy gait, as well as sports biomechanics [3,4] and recovery in gait impaired populations [1].

Unfortunately, the reliable estimation of GRF requires expensive measurement setups such as force plates. These may be installed under the floor or incorporated into treadmills. In either case, they either measure limited strides or restrict the movement space of the subject. It is therefore useful to explore wearable setups that allow freedom of movement, while providing reliable estimates of the GRF during gait or variable walking. Wearable alternatives [5] to these restricted laboratory setups include systems such as ForceShoes^TM^ [6] and pressure insoles [7], although each of them are associated with their respective drawbacks. ForceShoes^TM^ are bulky [8], and pressure insoles require additional analytical or machine-learning-based models to extract the 3D GRF [7,9].

Assuming a simple inverted pendulum gait model, GRF can be considered equal and opposite to the weight plus mass times linear accelerations at the CoM (CoM_acc_) [10], given no additional external forces are present. Therefore, if we can measure the CoM_acc_, we can estimate the GRF. Note that here, the GRF are the sum of all forces acting on the body, which is the sum of reactive forces at both feet, provided no additional contact with the environment. As inertial measurement units (IMU) measure accelerations of the segment they are attached to, the GRF acting on the body can be estimated either using a biomechanical model [11,12] or machine learning techniques [4,13,14]. Ancillao et al. [13] summarize several of these methods and find that they either estimate only the vertical GRF using a minimal setup or estimate the shear forces using machine learning methods or an array of several IMUs. The drawback of using machine learning methods includes the need for a representative training dataset. A minimal set of IMUs, combined with biomechanical models of gait is, therefore, a preferred setup for ambulatory sensing of 3D GRF.

In a previous study, we estimated the instantaneous 3D GRF during over-ground gait using a pelvis IMU [12] expressed in a body-centric frame. We first identified the pelvis segment frame ($\psi_{p}$) using a bowing calibration method [15]. Assuming that the CoM is encompassed within the pelvis, an error extended Kalman filter (EEKF) was designed to track the change in orientation of the CoM_acc_ during each step. IMUs placed on either foot were used to detect gait events and, additionally, provide the body-centric frame of reference. The heading for the reference frame was estimated using the movement of the feet, thereby avoiding the use of magnetometers. This avoids the handling of distortions induced due to measurement of magnetic field [16]. The body-centric frame provides a first person perspective, irrespective of the measurement setup, and thereby a functional representation of the gait, unlike a fixed global frame [12,17]. The average error across all walking tasks was 6 ± 1% body weight (BW). In that study [12], although 3D GRF were estimated using the pelvis IMU, the estimation of the body-centric frame ($\psi_{cs}$) required the use of foot IMUs. An ideal next step would be to estimate a body-centric frame using the pelvis IMU instead of the foot IMUs, in order to enable a minimal wearable setup.

Therefore, in this study, our goal is to use a pelvis IMU to measure 3D GRF during over-ground gait and express it in a body-centric frame also defined using the pelvis IMU. Ergo, we first estimate 3D GRF using methods from the previous study [12], and additionally, detect gait events, and a body-centric reference frame using information from the pelvis IMU. Different methods can be employed to estimate the heading for a body-centric frame using a pelvis IMU. For instance, the heading of the frame could be defined along the average pelvis acceleration over a few steps. However, validating this approach with reference setups is non-trivial. Change in CoM position could be used to define the heading, but it requires deriving the position from pelvis accelerations while correcting for drift, thereby introducing additional complexities. In this study, the heading of the body-centric frame was estimated using the direction of the high frequency CoM velocity (CoM_vel_). This approach was easier to validate with reference setups and less complex compared to the other approaches. The estimations of 3D GRF were then validated using reference setups and results from literature.

## 2. Materials and Methods

In this section, we first show the estimation of gait events using a pelvis IMU in Section 2.1, following which, we define the body-centric reference frame in Section 2.2. Then, in Section 2.3 we summarize the method used to estimate the GRF from the pelvis IMU. In Section 2.4, we describe the measurement setup and then the experimental protocol in Section 2.5. Finally, Section 2.6 summarizes the analysis that will be performed on the data.

### 2.1. Initial Contact Detection

Studies have investigated heuristic approaches to detect gait events using only a pelvis IMU [18,19,20,21]. We employ a simple approach using the accelerations measured in the pelvis frame ($\psi_{p}$). The $\psi_{p}$ frame was estimated from a forward bowing calibration method [12]. A second order zero phase Butterworth filter of cut off 2 Hz was used to low pass filter the sensor accelerations, which were transformed to $\psi_{p}$. The magnitude of the resulting signal was de-trended, and the peaks in the signal, which had a prominent height of at least 0.2 m/s^2^, were considered to be initial contact (IC) moments. Subsequently, we defined a step as the instance between two ICs.

### 2.2. Reference Frames Used

Here, we describe the definition of the body-centric frame. This frame is referred to in this study as the initial contact (IC) frame $\psi_{ic}$. The $\psi_{ic}$ frame is similar to the current step frame $\psi_{cs}$ [12], but relies only on information from the pelvis IMU. Figure 1 graphically defines the IC frame $\psi_{ic}$. The heading of the $\psi_{ic}$ frame was defined using the direction of CoM_vel_ estimated from the pelvis IMU during a step. The X axis of this frame is positive in the forward direction, and the Z axis lies along the vertical. The $\psi_{ic}$ was redefined per step, and the 3D GRF were transformed to this frame.

The steps required to estimate the $\psi_{ic}$ are shown in Figure 2. Table 1 lists the different notations employed in this study. The pelvis IMU measures accelerations ($y_{A}^{s}$) and angular velocities ($y_{G}^{s}$) in the sensor frame $\psi_{s}$. As mentioned earlier, the data were transformed to the pelvis frame $\psi_{p}$, which was defined using the bowing calibration method [12]. Then, during step *k*, an EEKF [12] was designed to track the change in orientation ($R_{i}^{ic\left(k-1\right),p}$) of the pelvis with respect to a predefined frame $\psi_{ic\left(k-1\right)}$ for a given sample *i*. This EEKF is described in detail in Mohamed Refai et al. [12]. The states tracked by the filter were orientation error $\theta_{\epsilon }$ and gyroscope bias error $b_{\epsilon }$. The change in orientation was first tracked with respect to a previous step *k* − 1, and then, using the change in orientation in step *k*, the current IC frame $\psi_{ic\left(k\right)}$ was estimated. The orientation estimated by integrating the angular velocity was corrected by inclination information derived from the accelerometer. The $R_{i}^{ic\left(k-1\right),p}$ was thus estimated using [22]:(1)$${\hat{R}}_{i}^{ic\left(k-1\right),p}={\hat{R}}_{i}^{ic\left(k-1\right),p,-}\left(I-{\tilde{\theta }}_{\epsilon }\right)$$

We assumed the initial orientation error ${\hat{\theta }}_{\epsilon ,init}$ to be zero. The initial gyroscope bias error ${\hat{b}}_{\epsilon ,init}$ was measured from gyroscope data when the subjects were standing still. Note that $R^{ic\left(k-1\right),p}$ is known at the beginning of each step *k*, as it would have been estimated using the EEKF in the previous step. However, an estimate of $R_{init}^{ic\left(k-1\right),p}$ is needed for the first step ever made. For this, the EEKF is run once for a few steps with an arbitrary initial heading estimate. After this, the change in orientation was used to estimate $R_{init}^{ic\left(k-1\right),p}$ [12,22].

In the following, we describe how the $\psi_{ic}$ frame is defined for each step. Pelvis accelerations in each step are now expressed in $\psi_{ic\left(k-1\right)}$(Equation (2)) as $y_{A}^{ic\left(k-1\right)}$ and must be transformed to $\psi_{ic\left(k\right)}$. Using arbitrary initial and final conditions, $y_{A}^{ic\left(k-1\right)}$ was high-pass filtered using a second order zero phase Butterworth filter with a cut off of 2 Hz to obtain the high frequency CoM_vel_. Then, during step *k*, the time instance *m* was selected when the magnitude of CoM_vel_ vector was highest in the XY plane. At this time, instance *m*, the direction of the velocity vector in the XY plane, was defined using Equation (3) below. This was the heading or X axis for $\psi_{ic\left(k\right)}$. After assuming that the Z axis lay along the vertical (Equation (4)), the $R_{k}^{ic\left(k\right),ic\left(k-1\right)}$ was determined (Equation (5)). This was redefined for each step, resulting in a $\psi_{ic\left(step\right)}$ per step. Note that, in this study, a step is the instance between subsequent ICs.
(2)$$y_{A,i}^{ic\left(k-1\right)}={\hat{R}}_{i}^{ic\left(k-1\right),p}\cdot y_{A,i}^{p}$$
(3)$$X=\frac{CoM_{vel,m}}{\left\|CoM_{vel,m}\right\|}$$
(4)$$Z={\left[0\;0\;1\right]}^{T}$$
(5)$$R_{k}^{ic\left(k\right),ic\left(k-1\right)}=\left[\begin{array}{ccc} X & Z\times X & Z \end{array}\right]$$

### 2.3. Estimating Ground Reaction Forces

The accelerations $y_{A}^{ic\left(k-1\right)}$ in frame $\psi_{ic\left(k-1\right)}$ (Equation (2)) were transformed to the frame $\psi_{ic\left(k\right)}$ using $R_{k}^{ic\left(k\right),ic\left(k-1\right)}$ per step *k* (Equation (6)). As we assume the pelvis accelerations to be similar to CoM_acc_, the GRF (**GRF**_IM_) were estimated using Newton’s second law (Equation (7)).
(6)$$y_{A,i}^{ic\left(k\right)}=R_{k}^{ic\left(k\right),ic\left(k-1\right)}\cdot y_{A,i}^{ic\left(k-1\right)}$$
(7)$$GRF_{A,i}^{ic\left(k\right)}=mass\cdot y_{A,i}^{ic\left(k\right)}$$

During preliminary analysis, we identified sharp peaks around the IC instances, possibly due to impact in the estimated 3D GRF. An adaptive peak removal algorithm was employed to remove these peaks [12]. The peaks around an IC were first identified by detecting the local maxima and minima. Then, the signal in this region around the peak was smoothened using a Savitsky Golay smoothing filter [23] of order 3. Following this, a second order zero phase Butterworth band pass filter with a cut off range of 0.1–5 Hz and 0.1–3 Hz was used to filter the X and Y axis, respectively. For the Z axis, a second order zero phase Butterworth low pass filter with a cut off of 10 Hz was employed.

### 2.4. Measurement System

Figure 3a shows the sensor setup; a single Xsens^TM^ MTw IMU was placed at the lower back on the pelvis. The data from the IMU were read using an MT Manager (version 4.8) software (Xsens^TM^, Enschede, The Netherlands) at 100 Hz. We employed two reference systems in this study. The ForceShoe^TM^ (Xsens^TM^, Enschede, The Netherlands), consisting of two 6DoF force and torque sensors per foot, was used for validating the estimation of GRF. IC instances were determined when the magnitude of GRF on each foot exceeded 30 N. The GRF on both feet were summed to obtain the total reference GRF (**GRF**_FS_), which is equal and opposite to the body weight plus mass times CoM_acc_ [6].

The frame $\psi_{ic}$ for our IMU-based system was defined using equations (2–5). Similarly, we need to determine the frame $\psi_{ic}$ for the reference datasets. For this purpose, we measured the kinematics of CoM using a VICON© (Oxford Metrics PLC., Oxford, UK) motion capture system. Markers were placed on the right anterior superior iliac spine, right posterior iliac spine, left anterior superior iliac spine, and left posterior iliac spine. We assumed that the position of the CoM was at the centroid of the pelvis, demarcated by the four pelvis markers. Velocities and accelerations of the CoM were obtained using differentiation and subsequent low pass filtering with a second order zero phase Butterworth filter of cut off 10 Hz. Then, gravitational acceleration was added to the Z axis of the accelerations to obtain the CoM_acc_. A second order zero phase Butterworth high-pass filter with cut off of 2 Hz was used to obtain the CoM_vel_. The direction of the velocity vector in the XY plane was used to transform the reference **GRF**_FS_ to the frame $\psi_{ic}$ using the steps defined in Section 2.2. Thus, we estimate the acceleration from the VICON© position data, and then integrate it after including gravitational acceleration to obtain the high frequency CoM_vel_, in order to make sure that our reference $\psi_{ic}$ frame was estimated in a similar manner as the IMU-based system.

The data from VICON© and ForceShoe^TM^ were sampled at 100 Hz. The data from Xsens^TM^ IMU, ForceShoe^TM^, and VICON© were synchronized. The subjects raised their right foot before each task, and this movement was used for the synchronization of the three systems.

### 2.5. Participants and Experimental Protocol

Three healthy male subjects were recruited for the study. The average height, weight, and age was 1.8 ± 0.04 m, 74.3 ± 7.6 kg, and 25.6 ± 3.3 years, respectively. Before the experiment, each participant signed an informed consent. The study was conducted in accordance with the Declaration of Helsinki, and the protocol was approved by the Ethical Committee of the research faculty under protocol number RP 2019-108.

The experimental protocol is shown in Figure 3b [24]. The subjects began by standing still for a few seconds, following which they were asked to bend the trunk forward thrice. This movement is used for the calibration. Once the researcher gave the start sign, the subject performed each of the following walking tasks four times:Normal Walk (NW): the subjects walked at their preferred walking speed for 5 m.L Walk (LW): the subjects walked for 3 m and then turned 90° to the right, and walked for 2 m.Walk and Turn (WT): the subjects walked for 5 m, and then turned and walked back to start position.Walk and Turn Twice (WT2): the subjects walked for 5 m, turned and walked back to start position, and then turned again and walked for 5 m.Slalom Walk (SlW): the subjects walked in a snake-like slalom pattern. A pylon was placed at 2 m and another at 4 m to help them with this pattern.

### 2.6. Analysis of Results

First, we validated the estimation of IC instances using the information from the pelvis IMU. Then, we evaluated the differences in heading ($\theta_{d}$) between the $\psi_{ic}$ frames defined using the pelvis IMU, and that of the reference setup. This was estimated by measuring the angle between the heading vectors used to define the $\psi_{ic}$ frames. Following this, we test the accuracy of our method in estimating 3D GRF using different analyses. This includes measuring the root mean square of the differences (*RMS*), and Pearson’s correlations (*CORR*) between the estimated 3D GRF (**GRF**_IM_) from the pelvis IMU and the reference **GRF**_FS_ for the different walking tasks. A Bland–Altman analysis was also performed. MATLAB^®^ 2018b (MathWorks, Natick, MA, USA) was used for all analyses.

## 3. Results

Some trials had to be excluded from the analysis due to technical issues with the reference system. However, it was made sure that each subject had at least three walking trials per task.

Figure 4 compares the **GRF**_IM_ shown in blue and **GRF**_FS_ shown in red, for a subject performing a WT trial. The difference between them for each axis is shown in black. The subject makes a 180° turn at 25 s, highlighted by a red area in the graphs.

Table 2 summarizes the results of the analysis and compares the method against reference setups for each walking task. First, the average mean error in estimating IC instances per task is summarized in the column **IC**. Based on preliminary comparison with reference values, the estimated IC instances were adjusted for a uniform offset of 0.08 s for all trials. Using our simplified approach, the average median error in estimating IC was found to be 2 ± 4.4 ms across all walking tasks. Table 2 also shows the average heading error $\theta_{d}$ for the $\psi_{ic}$ frames, excluding the first and last steps made. We see that the NW task has the highest errors with respect to estimation of IC, and therefore, the $\theta_{d}$, as the $\psi_{ic}$ frames are identified between ICs. Then, Table 2 summarizes the errors in estimating the 3D GRF over the complete gait, including quiet standing, gait initiation, turning events, and termination. The *RMS* values shown in the table are an average of all trials of all subjects for each walking task. The maximum *RMS* across all axes was found to be 5.7, 6, 6.8, 7.2, and 5.8% BW for the NW, LW, WT, WT2, and SlW walking tasks, respectively. The average *RMS* of the magnitude of the GRF was 5 ± 0.4% BW across all walking tasks. The WT2 task showed a slightly larger error across the XY plane, probably because it had more changes in heading. The errors normalized against the range of the reference GRF values (*NRMSE*) were found to be 16.3 ± 1.7% across all walking tasks. We found an average *CORR* of 0.5 ± 0.2 for the shear GRF, and a higher correlation of 0.8 ± 0.03 in estimating vertical GRF. We estimated the *RMS* and correlation between the measurements for the complete gait cycle.

Figure 5 shows the Bland–Altman plot comparing the magnitude of the estimated shear GRF (GRF in the XY plane) from **GRF**_IM_ with the reference **GRF**_FS_. The values for magnitude of shear GRF were not normally distributed. Therefore, the mean difference is shown along with the interquartile ranges (IQR) in the figure. The mean difference between the systems is on average 0.24% BW across all tasks. We see a concentration of differences for mean shear GRF values close to 0% BW. The difference between the systems becomes more random as the mean magnitude of shear GRF is larger. Figure 6 depicts the Bland–Altman comparison for the estimation of vertical GRF. Here, the average of the mean difference across all walking tasks was found to be −2.7% BW. In this figure, we find a concentration of the difference spread across the vertical GRF close to 100% BW. For other values of vertical GRF, the difference is spread randomly. Note that 0% BW and 100% BW are the GRF values during no-motion for the shear GRF and vertical GRF, respectively. Hence, they show larger concentrations of the difference between systems.

## 4. Discussion

The methods used in this study to estimate 3D GRF are similar to our previous study [12]. Here, we additionally describe how to estimate gait events, and the body-centric initial contact frame using the pelvis IMU, thereby avoiding the need for foot IMUs. This enables development of a minimal sensing system for 3D GRF during gait. The method has been applied to a limited set of subjects, but a range of variable walking tasks, and shows the estimation of 3D GRF during the complete gait trial, including gait initiation, walking, and termination.

A number of assumptions have been made in this study. We assume an inverted pendulum model of gait, where the CoM is the swinging bob. We also assumed that the CoM moves within the pelvis, and that the accelerations can be measured with a pelvis IMU. The GRF opposes gravity and accelerates the CoM. Our methods are restricted to situations when only the feet are in contact with the environment. Our methods estimate the total GRF acting on the body, and therefore, we do not measure how the weight shifts from one foot to another.

The only gait events estimated were the IC instances. There are several methods in literature for estimating ICs using one pelvis IMU [21]. Our average median error in estimating IC was found to be similar to the results found using the method of Lee et al. [19,21], which was 2 ms. Although we found one walking trial where our IMU-based method estimated an additional IC instance, our method is much simpler to that of Lee et al. [19]. Our largest error was found for the *NW* task, which was 20 ms. Nonetheless, the robustness of IC detection may be improved with alternatives in literature [21]. Our algorithm does not differentiate between left and right ICs, as that information is not required in this study. Differentiating left and right gait events from pelvis IMU data is challenging, but possible [21]. The estimation of the heading for the $\psi_{ic}$ frame may be further improved with this knowledge, especially when measuring asymmetrical gait such as hemiparesis after stroke. During asymmetrical gait, it might be necessary to distinguish turning from asymmetrical inclination of the body towards the less affected side, while defining the heading for the for the $\psi_{ic}$ frame. In Table 2, we find larger errors in the heading ($\theta_{d}$) for the NW task. This could be influenced by the larger mismatch in IC instances, which in turn, has an influence on the selection of time window for the steps. The $\psi_{ic}$ frame is a reference frame attached to the body, thereby tracking its kinetics irrespective of the change in direction. The use of such a reference frame avoids the need to correct for drift with respect to a fixed global frame. Our magnetometer free approach is insensitive to magnetic disturbances, which is an additional advantage.

Ancillao et al. [13] identify that the most challenging task when using IMUs to estimate GRF is determining the shear GRF; the antero-posterior and medio-lateral components. As we assume that the CoM is located within the pelvis, its accelerations are estimated using the pelvis IMU. The estimation of CoM_acc_ from the pelvis IMU in the antero-posterior, medio-lateral, and vertical axes by the EEKF serves as the largest influence of errors. The estimation of CoM_acc_ could be improved using additional biomechanical models [25]. Nevertheless, our results show that it is possible to estimate GRF using a single pelvis IMU. For instance, in Figure 4 we see overlap between **GRF**_IM_ and **GRF**_FS_ for the complete gait cycle. Table 2 summarizes the errors in estimating the 3D GRF for the complete walking tasks, from start to stop. The WT2 task showed a slightly larger error across the XY plane, probably because it had more changes in heading. The average *NRMSE* of 16.3 ± 1.7% for all walking tasks is slightly larger than our previous study [12], where we found an average *NRMSE* of 12.1 ± 3.3%, and also that of Leporace et al. [26] who found an average of 9.3 ± 6.4% in the horizontal plane. The **GRF**_IM_ correlated strongly with the reference in the vertical axis due to the large influence of gravity and correlated weakly in the Y axis because of larger errors in this axis. We found all *CORR* to be significant (*p* < 0.01). Our average *CORR* for the vertical GRF is close to the results of Jiang et al. [27], in which an array of IMU sensors were used to estimate only the vertical GRF, with an average *RMS* of 2% BW, and high correlations of 1. Nevertheless, our method offers an estimation of 3D GRF albeit with slightly larger errors.

The low number of subjects and the low variability in age and gender are limitations. Our calibration method requires a bowing movement, which might be difficult for subjects with back issues. Nevertheless, this paper presents a new method to estimate the 3D GRF as a function of time in a body-centric frame employing a single pelvis IMU, and thus offers a proof of principle of this new method. Finally, using simple models [11,28], and knowledge of distinct left and right gait events [21], the 3D GRF may also be separated into GRF acting on either foot.

## 5. Conclusions

The study shows the feasibility of using a single pelvis IMU to track the 3D GRF during over-ground gait and expressing it in a body-centric initial contact reference frame. The shear GRF were estimated with a root mean square error of 5.2 ± 0.5% BW over the complete gait cycle including initiation and termination of gait. Though these margins are comparable with the literature, further validation studies in which more subjects, including those with gait impairment, are required. Furthermore, more variable walking must be studied.

## Figures and Tables

**Figure 1 sensors-20-06363-f001:**
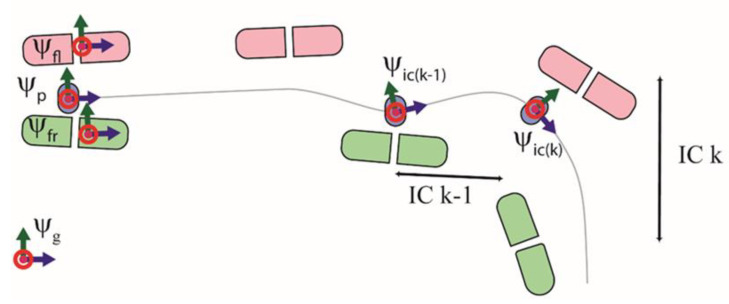
Graphical interpretation of the reference frames. The left foot is in pink, and the center of mass (CoM) trajectory is the thin gray line. Instead of a fixed global frame $\psi_{g}$, an initial contact frame $\psi_{ic\left(k\right)}$, which depends on the direction of the CoM velocity vector in step *k*, is employed.

**Figure 2 sensors-20-06363-f002:**
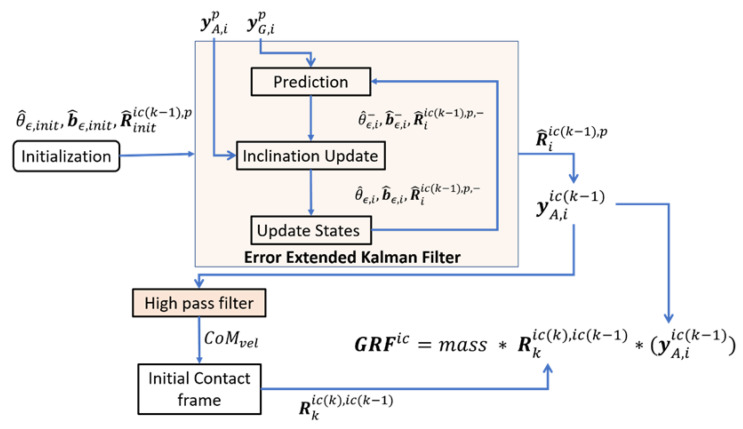
Overview of the method used to estimated 3D ground reaction forces (GRF): the error extended Kalman filter [12] tracks the orientation error $\theta_{\epsilon }$ and gyroscope bias error $b_{\epsilon }$, to estimate the $R_{i}^{ic\left(k-1\right),p}$ for each step. Then, $R_{step}^{ic\left(k\right),ic\left(k-1\right)}$ was estimated using the direction of CoM velocity.

**Figure 3 sensors-20-06363-f003:**
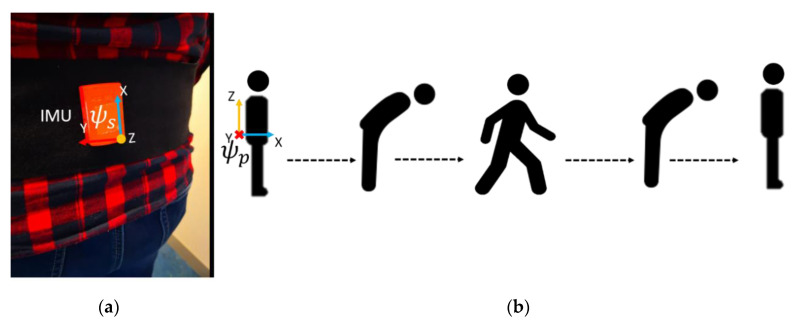
(**a**) Placement of the Xsens^TM^ MTw inertial measurement unit (IMU) at the lower back of the subject. The sensor frame $\psi_{s}$ of the IMU is also shown. (**b**) A simplified overview of the experimental protocol. The subjects stand still for a few seconds, following which they bow thrice, and then perform the walking task. After this, they bow again and stand still for a few seconds before the measurement is stopped. The bowing movement is used to determine the pelvis frame $\psi_{p}$ seen in the figure.

**Figure 4 sensors-20-06363-f004:**
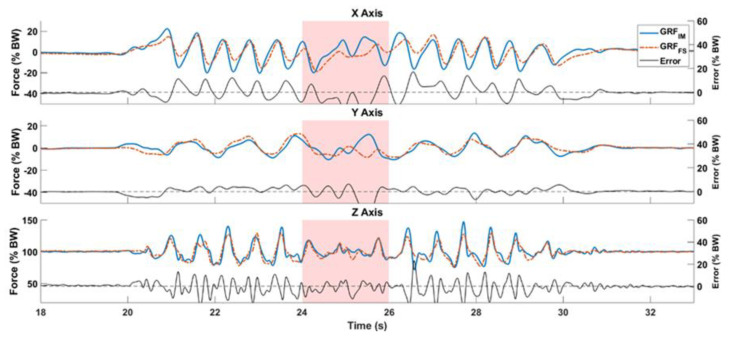
Estimated and reference GRF compared for a Walk and Turn (WT) task. The subject makes a 180° around 25 s highlighted with the shaded region.

**Figure 5 sensors-20-06363-f005:**
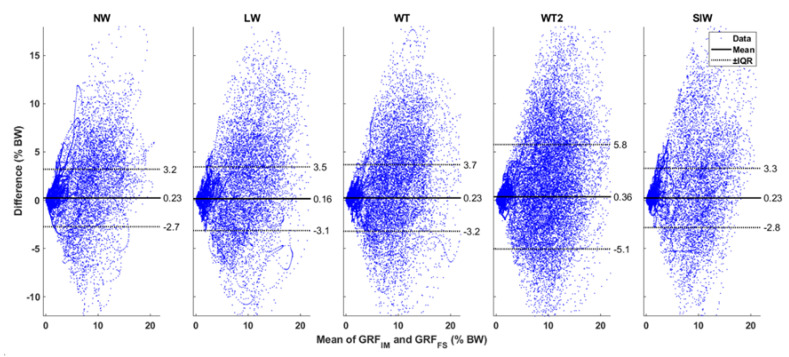
Bland–Altman plots: The magnitude of the shear GRF are compared between the reference GRF_FS_ and estimated GRF_IM_. The mean shear GRF of the two systems are plotted along the X axis, and the difference between them is shown along the Y axis. All data are in % body weight. The mean of the differences is shown by a thick black line. The data were not normally distributed, and the interquartile ranges (IQR) are shown by dotted black lines. The values of mean and the IQR are also shown in the graph. NW: Normal Walk, LW: L Walk, WT: Walk and Turn, WT2: Walk and Turn Twice, SlW: Slalom Walk.

**Figure 6 sensors-20-06363-f006:**
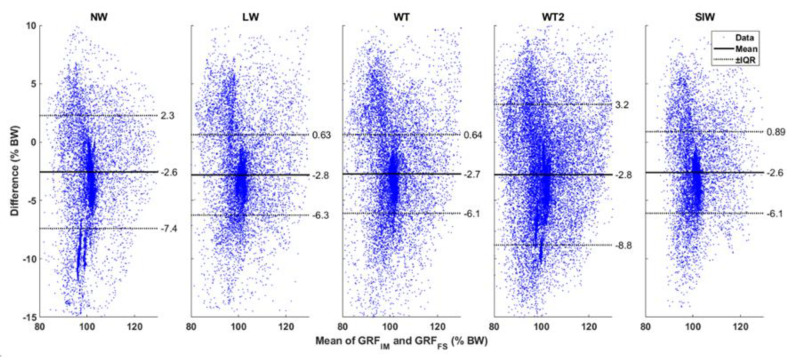
Bland–Altman plots: The magnitude of the vertical GRF are compared between the reference GRF_FS_ and estimated GRF_IM_. The mean vertical GRF of the two systems are plotted along the X axis, and the difference between them is shown along the Y axis. All data are in % body weight. The mean of the differences is shown by a thick black line. The data were not normally distributed, and the interquartile ranges (IQR) are shown by dotted black lines. The values of mean and the IQR are also shown in the graph. NW: Normal Walk, LW: L Walk, WT: Walk and Turn, WT2: Walk and Turn Twice, SlW: Slalom Walk.

**Table 1 sensors-20-06363-t001:** Notations used, shown for an arbitrary vector ***a***.

Notation	Definition
$a_{k}$	***a*** at k-th instant
$a^{s}$	***a*** expressed in frame $\psi_{s}$
$\dot{a}$	derivative of ***a***
$\hat{a}$	a posteriori estimate of ***a***
$a^{-}$	a priori estimate of ***a***
$\tilde{a}$	skew symmetric operator on ***a***
$e_{a}$	Gaussian white noise associated with ***a***

**Table 2 sensors-20-06363-t002:** Differences between IMU-based **GRF**_IM_ and reference ForceShoe^TM^-based **GRF**_FS_: initial contact (IC) estimation, heading differences ($\theta_{d}$), root mean square of the differences (*RMS*), and Pearson’s correlations (*CORR*).

---	IC (ms)	$\theta_{d}$ (deg)	RMS_X_ (%)	RMS_Y_ (%)	RMS_Z_ (%)	CORR_X_	CORR_Y_	CORR_Z_
**NW**	20	18.33 ± 8.57	4.47 ± 1.42	4.38 ± 1.17	5.14 ± 0.89	0.68 ± 0.24 *	0.29 ± 0.21 *	0.73 ± 0.11 *
**LW**	0.21	16.89 ± 9.52	5.42 ± 1.35	4.86 ± 1.04	5.02 ± 0.85	0.66 ± 0.14 *	0.40 ± 0.14 *	0.81 ± 0.06 *
**WT**	1.35	13.11 ± 9.60	5.55 ± 1.50	4.72 ± 2.16	5.46 ± 1.46	0.64 ± 0.25 *	0.46 ± 0.34 *	0.79 ± 0.05 *
**WT2**	2.12	12.97 ± 6.96	6.92 ± 1.89	5.02 ± 1.33	5.54 ± 0.84	0.61 ± 0.19 *	0.50 ± 0.16 *	0.82 ± 0.04 *
**SlW**	−0.49	7.83 ± 6.47	5.50 ± 0.94	5.00 ± 0.57	4.38 ± 1.13	0.56 ± 0.08 *	0.43 ± 0.05 *	0.81 ± 0.04 *

All values are an average of the three subjects that were tested. The RMS is expressed in % body weight. NW: Normal Walk, LW: L Walk, WT: Walk and Turn, WT2: Walk and Turn Twice, SlW: Slalom Walk. * All correlations were found to be significant (*p* < 0.01).

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
