# Peer review of "Portable Gait Lab: Estimating Over-Ground 3D Ground Reaction Forces Using Only a Pelvis IMU"

_sensors, 2020, doi:10.3390/s20216363_

Round 1

Reviewer 1 Report

This study is meaningful comparing the IMU sensor on pelvic and ground reaction force. However, I think the following points are insufficient and should be supplemented. 

Line182  Sample size

The numbers of subjects in this study are only three healthy persons. It is a very low sample size. The purpose of this study is to validate the accuracy of an IMU sensor compare with 3D GRF. Is this enough for three people to check validation? Sample sizes may be evaluated by the quality of the resulting estimates. 

Line226 correlations analysis

In general, Intraclass coefficient correlation(ICC) analysis is used to check the accuracy of the sensor. The correlation analysis is also a method to check the consistency. However, I think the correlation analysis only is not enough to explain the validity of the sensor. 

Line166 XYZ deviation of IMU sensor and ground reaction force

Could the movement of an IMU sensor on pelvic be called GRF? Can you explain the difference when the weight on the foot changes in a similar movement in your study?

Line185 protocol number of Ethical committee

Please, write your protocol number in your manuscript.

Line 226 little information about table 2

Please, write the following information

subject number, information of the compared device. ex) IM, FS

Please write down to the second decimal place. 

Additionally, add a figure about bland Altman plot

The bland Altman plot is described as the agreement between two quantitative measurements. 

Author Response

General comment: Thank you very much for your review. Each comment has been responded to, and the changes are addressed in the paper. Please note that we found some issues with our synchronization methods, and therefore improved it. The results have been updated accordingly.

This study is meaningful comparing the IMU sensor on pelvic and ground reaction force. However, I think the following points are insufficient and should be supplemented. 

 Line182  Sample size

The numbers of subjects in this study are only three healthy persons. It is a very low sample size. The purpose of this study is to validate the accuracy of an IMU sensor compare with 3D GRF. Is this enough for three people to check validation? Sample sizes may be evaluated by the quality of the resulting estimates. 

  • Indeed the sample sizes are limited in this case. Further analysis must be performed and the number of subjects is a limitation of this study. Nevertheless, the study presents a new method and also offers proof of principle of the method. The differences with a reference system are studied using different tools, including root mean square of the differences, correlation analysis, and Bland-Altman analysis.
  • The improvement of synchronization has improved our correlations slightly for all axes. The results have been updated.
  • Changes have been made to the text in line with this comment. Please refer to the updated Table 2, Figures 5 and 6, Lines 252 – 263, 287, and 332.

 Line226 correlations analysis

In general, Intraclass coefficient correlation(ICC) analysis is used to check the accuracy of the sensor. The correlation analysis is also a method to check the consistency. However, I think the correlation analysis only is not enough to explain the validity of the sensor. 

  • You are right. Correlation analyses are not enough. As mentioned earlier, we have also looked at a number of other analyses. In addition to measuring the RMS and Pearson’s correlations, we also tested the significance of the correlations, included Bland-Altman plots according to your suggestions, and studied closely the differences in heading estimation for each step.
  • Changes have been made to the text in line with this comment. Please refer to the lines 214.  

Line166 XYZ deviation of IMU sensor and ground reaction force

Could the movement of an IMU sensor on pelvic be called GRF? Can you explain the difference when the weight on the foot changes in a similar movement in your study?

  • Our methods are based on a few principles. One is that IMU measures accelerations, which can be converted to forces using Newton’s second law. Second, we know that center of mass accelerations are equal and opposite the total body ground reaction forces. In our study, we assumed that the centre of mass is located close to the pelvis, and therefore, and IMU placed in this region could measure the accelerations at the pelvis. The main challenge is to identify the accelerations in the Anterio-posterior, medio-lateral, and vertical directions. The forces extracted from the AP and ML accelerations contribute to the shear forces acting on the body while walking.
  • The GRF in this study is not the GRF acting under each foot, but a sum of GRF acting at both feet. The assumptions regarding measuring GRF using pelvis IMU is valid under the given assumptions.
  • Changes have been made to the text in line with this comment. Please refer to lines 38-42, and 291 – 296.

Line185 protocol number of Ethical committee

Please, write your protocol number in your manuscript.

  • Included in Line 196.

Line 226 little information about table 2. Please, write the following information. subject number, information of the compared device. ex) IM, FS. Please write down to the second decimal place. 

  • The required information has been added to Table 2

Additionally, add a figure about bland Altman plot

The bland Altman plot is described as the agreement between two quantitative measurements. 

  • Please see figures 5 and 6, and their corresponding text in Lines 253 – 264.

Reviewer 2 Report

The paper describes an original methodology for estimating ground reaction force acting at the center of mass level, using only an IMU. The paper is surely of interest, although being probably unorthodox. As the ground force is just modelled as mass times acceleration, assuming the mass as known, basically authors are measuring the components of acceleration. The measure of GRF might sound a little pretentious. However, I think that the manuscript could gain value and deserve publication if they clarify a series of issues:

  1. How is the procedure actually validated? This is not completely clear. It seems that authors measured force in the ForceShoe reference frame and then rotated using the same rotation matrix calculated by their algorithm. I would have used the rotation matrix extracted from VICON data to obtain reference signal. In case they did it, please clarify.
  2. A key improvement of the manuscript with respect to a previous work is the use of a single IMU to detect initial contact. What is the accuracy of such a methodology? I think this is crucial but the paper seems to be rushed about it. Please report it.
  3. Authors reported the average value of RMS error. What do they intend as average? Averaged on the different trials? By looking at the graphs reported the error seems to suggest a higher value. Do they calculated RMS also including periods of no-motion? I think that also the maximum RMS value should be reported to give to the reader a clear feeling about the possible use of this methodology.

Author Response

General comment: Thank you very much for your review. Each comment has been responded to, and the changes are addressed in the paper. Please note that we found some issues with our synchronization methods, and therefore improved it. The results have been updated accordingly.

The paper describes an original methodology for estimating ground reaction force acting at the center of mass level, using only an IMU. The paper is surely of interest, although being probably unorthodox. As the ground force is just modelled as mass times acceleration, assuming the mass as known, basically authors are measuring the components of acceleration. The measure of GRF might sound a little pretentious. However, I think that the manuscript could gain value and deserve publication if they clarify a series of issues:

  1. How is the procedure actually validated? This is not completely clear. It seems that authors measured force in the ForceShoe reference frame and then rotated using the same rotation matrix calculated by their algorithm. I would have used the rotation matrix extracted from VICON data to obtain reference signal. In case they did it, please clarify.
    1. You are right. That is how we estimated the reference values. The reference force values were measured from the ForceShoe™. Then data from the VICON© measurements were used to define the body centric reference frames for each step. In order to make sure we are not comparing apples to pears, we extracted the heading of the CoM velocity from the VICON data and then used it to make the reference rotation matrices. The differences in the reference heading frames and that of the IMU derived frames are mentioned in Table 2.
    2. Changes have been made to the text in line with this comment. Please refer to the lines 174 – 176, and 185 – 187.
  2. A key improvement of the manuscript with respect to a previous work is the use of a single IMU to detect initial contact. What is the accuracy of such a methodology? I think this is crucial but the paper seems to be rushed about it. Please report it.
    1. The reliance of our methods in estimating initial contact is discussed further, and also reported in Table 2.
    2. Changes have been made to the text in line with this comment. Please refer to table 2, and the lines 210, 235-242, and 298-311.
  3. Authors reported the average value of RMS error. What do they intend as average? Averaged on the different trials? By looking at the graphs reported the error seems to suggest a higher value. Do they calculated RMS also including periods of no-motion? I think that also the maximum RMS value should be reported to give to the reader a clear feeling about the possible use of this methodology.
    1. The average RMS is estimated as an average of the values estimated across the subjects. Yes, we did measure the difference during periods of no-motion and also turns. We have reported the maximum RMS in the article (Line 245). They don’t seem to be very different from the average values.
    2. Changes have been made to the text in line with this comment. Please refer to changes made in line 215, and 242 – 246.

Round 2

Reviewer 1 Report

It has been appropriately corrected for the point out.

Reviewer 2 Report

Authors covered all my issues.